# Structural Flexibility in Triboelectric Nanogenerators: A Review on the Adaptive Design for Self-Powered Systems

**DOI:** 10.3390/mi13101586

**Published:** 2022-09-23

**Authors:** Zequan Zhao, Yin Lu, Yajun Mi, Jiajing Meng, Xia Cao, Ning Wang

**Affiliations:** 1Center for Green Innovation, School of Mathematics and Physics, University of Science and Technology Beijing, Beijing 100083, China; 2Beijing Institute of Nanoenergy and Nanosystems, Chinese Academy of Sciences, Beijing 100083, China; 3School of Chemistry and Biological Engineering, University of Science and Technology Beijing, Beijing 100083, China

**Keywords:** wearable electronics, TENG, self-powered system, IoT application

## Abstract

There is an increasing need for structural flexibility in self-powered wearable electronics and other Internet of Things (IoT), where adaptable triboelectric nanogenerators (TENGs) play a key role in realizing the true potential of IoT by endowing the latter with self-sustainability. Thus, in this review, the topic was restricted to the adaptive design of TENGs with structural flexibility that aims to promote the sustainable operation of various smart electronics. This review begins with an emphatical discussion of the concept of flexible electronics and TENGs, and continues with the introduction of TENG-based self-powered intelligent systems while placing the emphasis on self-powered flexible intelligent devices. Self-powered healthcare sensors, e-skins, and other intelligent wearable electronics with enhanced intelligence and efficiency in practical applications due to the integration with TENGs are illustrated, along with an emphasis on the design strategy of structural flexibility of TENGs and the associated integration schemes. This review aims to cover recent achievements in the field of self-powered systems, and provides information on how flexibility or adaptability in TENGs can be adopted, their types, and why they are required in promoting advanced IoT applications with sustainability and intelligence algorithms.

## 1. Introduction

Recently, flexible electronics have been attracting tremendous attention because of their merits, such as high compatibility, conformal contact with human skin, and easy integration on diverse IoT platforms. However, toward the rich application scenarios, for instance, personalized medicine [1,2], chemical sensors [3], personal motion monitoring [4], personalized electronic device customization [5], and so on, the true potential of IoT can only be realized if they are made self-sustainable, either by reducing the power consumption of the IoT through ultra-low-power-consumption circuits and low-voltage operation, or combining them with energy-harvesting technologies and making more energy available, or in most cases, hand in hand [6,7,8]. Meanwhile, as an emerging and ever-growing technological field, another challenging issue is to make flexible electronic materials as well as integrated power sources both durable and powerful in strained states, especially for applications such as skin-like electronics, implantable biodegradable devices [9,10,11], and bioinspired soft robotics [12,13,14,15,16]. Highly stretchable and biocompatible energy technologies will greatly boost the development of next-generation intelligent lifelike electronics [17,18].

Wang and his team reported in Science in 2001 that ZnO semiconductor material tape was synthesized for the first time. Then, using the piezoelectric and semiconductor properties of ZnO, a piezoelectric nanogenerator (PENG) with small volume and wide energy collection range was invented in 2006. Later, in order to further develop nanogenerators, improve energy collection range, and expand application scenarios of nanogenerators, Wang and his team proposed the triboelectric nanogenerator (TENG) in 2012. Since it was invented by Wang and his colleagues, the TENG has been proven as both a promising energy-converting technology for collecting mechanical energy from the ambient environment and a highly sensitive sensor for dynamic force detection [19,20,21]. Because of the extensibility, portability, low cost, and high compatibility, the TENG is finding wide applications in various self-powered systems, ranging from integrated displays to wearable real-time healthcare electronics [22]. Especially, with the advantage of great customizability, affordability, and portability, self-powered systems based on TENGs with high structural flexibility are growing in popularity for various applications. Scientists are now exploring the greater design options for surprising new purposes and improving old devices with the key design considerations in terms of energy-conversion efficiency, dynamic detection sensitivity, and operation sustainability, and searching for more universal strategies for engineering stretchable TENGs regardless of the material [23,24,25].

Considering the importance of adaptive structural design for fast progress in TENG-based self-powered systems, this paper reflects the latest research trends and presents efforts in the structural design of flexible TENGs and aims to improve and maintain the performance of self-powered electronics in various application scenarios. We explain, in turn, the geometric and structural designs introduced to achieve (a) high-performance energy conversion, (b) high sensitivity toward desired mechanical stimuli, (c) reliable output in harsh environments, and (d) multifunction of reliable and flexible electronics. Finally, a perspective on reliable and flexible self-powered electronic devices that are developed based on TENGs is also presented for suggesting next-generation research (Figure 1).

## 2. Recent Progress in Structural Design for Self-Powered Systems

### 2.1. Working Principle of TENG

In principle, TENGs scavenge mechanical energy and transform it into alternating current based on coupled triboelectric effect and electrostatic induction. For the triboelectric effect, because different atomic nuclei have different binding capacities for extranuclear electrons, when two objects rub against each other, atoms with weak binding capacity for extranuclear electrons will lose electrons, while electrons with strong binding capacity for extranuclear electrons will gain electrons [22]. Currently, TENGs can operate under various working modes, for example, vertical contact separation mode [26], lateral sliding mode [27], single-electrode mode [28], and freestanding triboelectric layer mode [29] (Figure 2). Among them, the vertical contact separation mode is the most basic one, where triboelectric layers are generally made of two different triboelectric materials and stacked side by side, and their respective backs are attached with current collectors. When triboelectric layers are in contact, the same number of charges with opposite signs are formed on the contact surface. When the triboelectric layers are separated from each other, an air gap is formed in the center, and a prompted potential difference is shaped between the two electrodes. A current is then generated at the load between the two electrodes. When the triboelectric layers are close to each other, the potential contrast shaped by the triboelectric charge vanishes, and the electrons return. The flat sliding mode is similar to the upward vertical contact separation mode. The relative displacement between the triboelectric layers becomes the horizontal direction and then slides periodically to produce an alternating output. The single-electrode mode uses the Earth as an electrode, which can freely collect mechanical energy and is now widely used. In the independent layer mode, two detached electrodes are joined to the rear of the dielectric layer. At the point when the charged item reciprocates between the two electrodes, the potential difference between the two terminals changes and afterward drives electrons to stream between the two electrodes by the outer circuit [30].

Theoretically, the triboelectrification effect occurs when two materials with different triboelectric polarities touch, where electrons flow from the triboelectric positive surface to the negative one. When they are separated, a dipole moment arises to drive electrons through the external load, and the TENG acts as a charge pump, allowing current to flow back and forth between the electrodes in the form of alternating current [31], thus enabling a wide selection of electrode materials, including materials with high flexibility. It means that the structure of the TENG can develop as indicated by the required performance when there is a change from the already established state, and the primary adaptability of the TENG can even be developed as elastic and intelligent and may influence the operating mode of existing wearable electronics and the possible design of newly proposed self-powered devices. 

### 2.2. Structural Flexibility for High-Performance Energy Conversion

As mentioned above, the TENG is now considered as a new energy-harvesting technology and can collect energy from the ambient environment on the basis of coupled triboelectrification effect and electrostatic induction [32]. However, to continually and stably power various electronic devices, TENGs must be cost-effective and sustainable while offering features such as lightweight, small size, and high efficiency, even at low frequencies [33,34].

#### 2.2.1. Design of Rich Energy Collection Channels

To improve its output power, Wang’s team developed a new TENG in view of a conductive elastic sponge (ES-TENG) [35]. This particular structure has a great level of flexibility, and the adaptive deformation of the sponge allows it to collect kinetic energy of tumbling movements of different amplitudes from different flexible article surfaces, thus effectively improving the efficiency of energy harvesting (Figure 3a,b). Later, to enrich the application scenarios of e-textiles, Cao et al. reported a flexible and stretchable (Figure 3c) all-textile TENG that can collect energy from human movement, wind, and air noise [36]. The textile device is composed of three layers, one layer of silk fabric (positive electrode), one layer of conductive material (second electrode), and one layer of elastic conductive fabric woven from silicon-coated yarn (negative electrode and an electrode) (Figure 3d). The maximum power density of the fabric can reach 138.55 mW/m^2^ under the force of 50 N and 3 Hz, thus greatly widening its application scenarios.

#### 2.2.2. Increase Surface Charge Density

Generally, the triboelectrification-induced surface transfer charge density is now considered as the core factor that determines the output power of a TENG, whether the material is flexible or not [37,38]. Fan et al. precisely implanted N ions into polytetrafluoroethylene (PTFE) and polyethylene terephthalate (FEP) films to generate extremely negative triboelectric polymers (Figure 3e,f) with a surface charge density of four to eight times that of the original sample [39]. At the same time, the strong polarity greatly improved the dielectric constant of the material and further improved the energy storage density of the material. 

Increasing the charge density on the positive triboelectric film can also improve the output power. Sahoo et al. creatively introduced charge trapping layer (CTL) Al_2_O_3_ (Figure 3g) between the conducting electrode layer and positive triboelectric to construct an attractive and flexible graphene TENG for improving charge density [40]. In view of the trial results, after adding CTL, the output power of the advanced three-layer graphene TENG was 30 times higher than that of the TENG without CTL due to the synergistic effect between them.

#### 2.2.3. Design for Reducing Power Loss

The power output may be greatly affected if the conductivity of composite triboelectric electrodes is seriously degraded, elongated, or deformed, Zhong et al. proposed an omnidirectional stretchable triboelectric fabric based on liquid metal (LM) electrodes and Ecoflex (Figure 3h) [41]. The ultimate tensile strength of the triboelectric fabric is 660%, the resistance can be maintained at 0.678 Ω under 200% tensile strength, and the dielectric constant was improved by joining TiO_2_ particles into Ecoflex. The triboelectric fabric produced a maximum power density of 15.4 Wm^−2^ at 30 N and 3 Hz.

**Figure 3 micromachines-13-01586-f003:**
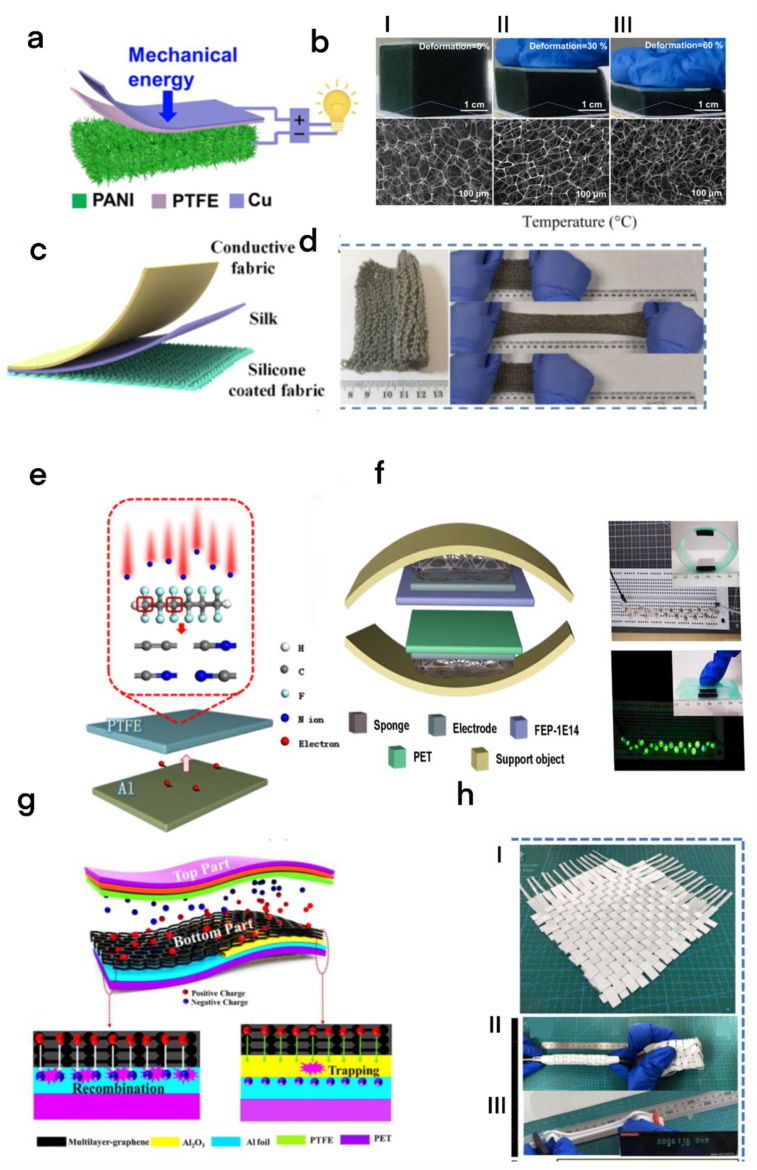
(**a**) Schematic representation of the ES-TENG. (**b**) (I–III) The optical and SEM pictures of a conductive flexible sponge with various compressive strains [35]. Liu et al. (2020), Elsevier. (**c**) Fabric TENG device structure for collecting acoustic energy. (**d**) Photos showing the elasticity of knitted fabrics [36]. Cao et al. (2022), Elsevier. (**e**) Schematic representation of TENG composed of Al and PTFE modified by ion implantation. (**f**) The left half is the schematic diagram of the equipment. The right half is the real device image based on the left, and 20 LED lights can light up in series [39]. Fan et al. (2021), Elsevier. (**g**) The schematic outline of the charge-catching system of 3L-Gr-TENG shows the case without and with the Al_2_O_3_ charge-catching layer [40]. Sahoo et al. (2021), MDPI. (**h**) (I) The finished product photo (II) two different deformation states of the OSHD-TENG. (III) LM electrode conductivity test photo [41]. Zhong et al. (2022), Elsevier.

### 2.3. Flexible TENG for Detection of Mechanical Stimuli

With the quick advancement of the Internet of Things and intelligent robots, information about most of the objects around us is expected to be collected in real time to the cloud through various possible networks [42]. Self-powered flexible sensors that are based on TENGs have gradually entered the historical arena. 

#### 2.3.1. Highly Customized Design

Due to the high sensitivity and extensibility of the TENG as a sensor, the TENG has been widely studied in the fields of real-time dynamic monitoring, intelligent robots, and personalized monitoring. Yang et al. used printed thermoplastic polyurethane as the triboelectric layer and elastic shell (Figure 4a) to produce a highly customized and flexible TENG (CF-TENG) [43]. Due to its high customization, it can focus on improving target sensing to achieve high accuracy that ordinary sensors cannot achieve. Thus, it can be effectively applied to fingers, wrists, or elbow joints to achieve a highly sensitive response.

#### 2.3.2. Multiple Components Designed for Coordinated Operation 

To realize the highly sensitive sensing, coordinated operation of multiple components is essential for TENG construction. Yuan et al. combined the sensor based on the TENG with the capacitive sensor to produce a multifunctional integrated sliding sensor [44], and gave an adaptable detecting plan to the delicate robot to accomplish high responsiveness and intelligent recognition (Figure 4b). Later, Zhu et al. developed an energy-self-sufficient, flexible multimode sensor system based on a thermoplastic polyurethane film [45]. It consisted of an intelligent conductive network composed of carbon ink and silver nanowire (C–Ag NW ink) through the screen-printing process (Figure 4f), and the system integrates a single-electrode TENG, flexible solid-state supercapacitor (FSSC), and strain sensor. The ingenious conductive network enables the strain sensor to recognize the strain applied, and the classic sandwich structure means that FSSC is used for energy storage. Due to the coordinated operation of various components, the as-constructed self-powered system has good detection and recognition ability for various physiological signals (expression, action, and voice) of the human body, and can be widely used in rehabilitation training, medical diagnosis, etc.

#### 2.3.3. Bionic Design and Improvement of Biocompatibility

To better simulate human skin and realize highly sensitive sensing, Zhao et al. designed a fingerprint-inspired electronic skin based on the TENG [46]. Through the bionic design of human fingerprint morphology (Figure 4c), it can respond to the fine texture (Figure 4d) and detect the contact area change caused by the dynamic contact between the bionic fingerprint structure and the measured object. The minimum size of the recognizable texture is as low as 6.5 µM and it can recognize disordered and ordered textures with an accuracy of 93.33% and 92.5%.

Compared to conventional sensors, flexible electronic skin can better fit the irregular human body and accept repeated bending and strain. Yu et al. designed a one contact separation TENG (CS-TENG) composed of ultra-flexible micro truncated array polydimethylsiloxane film and copper electrode (Figure 4e) [47], which can be well fitted on human skin to realize physiological signal observing with great sensitivity, stretchability, and short reaction time (60 ms).

#### 2.3.4. Highly Sensitive Textile Structure

Electronic textile products (e-textiles) have many advantages, such as flexibility, breathability, stretchability, and lightweight. Unlike electronic skin, which can only be used for part of the human skin (wrist and knee), electronic textiles can be easily integrated into clothes, shoes, and other products due to their excellent breathability and low weight. The electronic textile products based on TENGs can provide a feasible method for high-sensitivity sensing through the triboelectricity of TENGs [48,49,50].

Hao and his team proposed a highly integrated and expandable conductive composite fiber manufacturing process for weaving TENGs [51]. The fiber is formed by simultaneously injecting liquid alloy and silicon rubber into the separate entrance port of the coaxial needle, then automatically assembling from the exit end; this fiber has the structure of liquid alloy/silicone rubber core/sheath (Figure 4g). Due to the excellent flexibility and strain sensitivity of the fiber, it can be used as a strain sensor and can be made into T-TENG rings of different sizes to be worn on fingers, wrists, and elbows, or integrated into clothes, shoes, and other articles for monitoring their movement (Figure 4h).

**Figure 4 micromachines-13-01586-f004:**
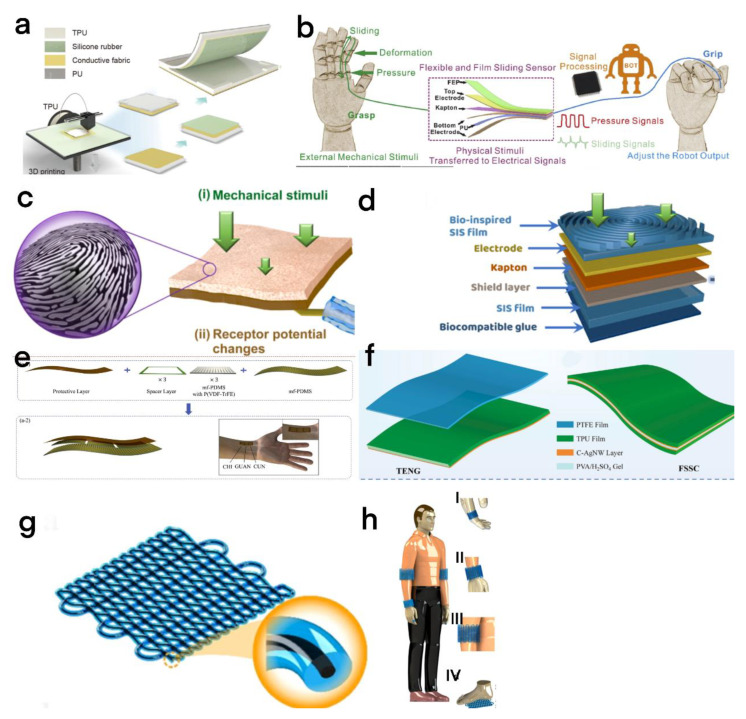
(**a**) The structure and manufacturing process of the CF-TENG and the optical picture of the insole sensing by pressure [43]. Yang et al. (2022), CNKI. (**b**) Schematic representation of the flexible sliding sensor. The device can be placed on the surface of a finger to collect pressure and glide signals [44]. Yuan et al. (2020), Elsevier. (**c**) Microstructure of human finger surface. (**d**) Design of a tactile sensing system based on Fe skin and artificial neural network [46]. Zhao et al. (2021), Elsevier. (**e**) Schematic diagram of CS-TENG manufacturing process and sensor array [47]. Yu et al. (2020), Elsevier. (**f**) Structural schematic of flexible TENG [45]. Zhu et al. (2022), Elsevier. (**g**) Schematic diagram of T-TENG based on a single fiber. (**h**) Wide application scenarios of T-TENG (I) a finger. (II) a wrist. (III) an elbow and (IV) a foot. [51]. Wu et al. (2022), Elsevier.

Yan et al. designed a wearable piezoresistive fiber-based triboelectric sensor (PRF-TES) utilizing nylon wires, silver nanowires, carbon nanotubes (CNTs), and PDMS as starting materials (Figure 5) [52]. Among them, nylon wires with a silver nanowire layer constitute flexible stretchable electrodes, and a piezoresistive CNTs/par film layer is used as a shell sensing element. It has good elasticity and flexibility and can be kept intact under 150% tension. It can be easily woven into large-area intelligent textiles for detecting human motion or posture. At the same time, because the two fibers have sensing ability, a sensor unit will be formed at their cross-contact point, so that it can map and quantify the static mechanical stress caused by pressure and strain, and sensitively detect the stress and strain itself, thus promoting the development of high-sensitivity and multimodal sensors, making them widely utilized in personal sports data monitoring.

### 2.4. Structural Design for Reliable Output in Extreme Conditions

Harsh environments such as acidic, alkaline, humid, and saline conditions are always challenges and barriers to the practical applications of TENG [53,54]. To maintain a stable output, the core part of the TENG, triboelectrode, as well as the friction layer, must be specially designed, either by encapsulation or adopting durable triboelectric materials that could be stable under a wide range of extreme conditions that are mentioned above.

#### 2.4.1. Corrosion Resistance

TENGs can continuously collect wave energy and provide power to sensors, which has great potential in ocean sensing. Ahn et al. designed a TENG (AR-TENG) based on a thermoplastic polymer with a nanopore pattern [55], which can maintain its performance after immersion in seawater or 1 million cycle tests, and its mechanical and chemical stability was not greatly affected. The highest peak output voltage reaches 360 V, and the current reaches 22 µA, which can provide sufficient power output for the ocean monitoring system for a long time.

Ye et al. reported a kind of full fabric TENG (F-TENG), which is composed of SiO_2_ nanoparticles and PVDF/perfluoroalkyl trichlorosilane (Figure 6a) [56], which not only has good air permeability, but also has excellent acid and alkali resistance and also shows hydrophobicity (static contact angle of 157°) and self-healing performance while providing effective power output under extreme conditions (Figure 6b).

#### 2.4.2. Wear Resistance

Wear-resistance-based long service life is another critical issue that should be addressed for practical application. Yang et al. designed a multifunctional single-electrode TENG (MF-TENG) [57], which consists of two self-healing silicone elastomer films and a thin self-healing polyvinyl alcohol-based hydrogel (Figure 6c) sandwiched between them. Due to the repairable network of dynamic imine bond in the charged layer and borate ester bond in the electrode, the original performance can be restored within 10 min after being damaged (Figure 6c), which greatly improves the stability and life of the TENG under extreme conditions.

**Figure 6 micromachines-13-01586-f006:**
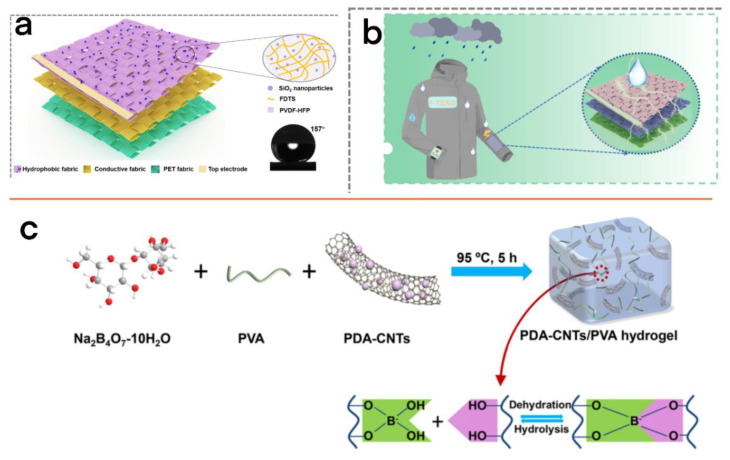
(**a**) Schematic representation of F-TENG. (**b**) The schematic graph shows the application situation of the wearable waterdrop energy collection sense [56]. Ye et al. (2021), ACS. (**c**) Preparation of PDA-CNT/PVA hydrogel and self-healing mechanism of PDA-CNT/PVA hydrogel [57]. Yang et al. (2021), ACS.

#### 2.4.3. Stretchability

Stretchability has always been an important advantage of flexible TENGs. Lu et al. developed one portable microstructured nano triboelectric generator (MS-TENG) [58]. It is composed of special microstructures which have PDMS (polydimethylsiloxane) film, lithium chloride polyacrylamide (LICLPAAM) hydrogel, and fluorinated ethylene propylene (FEP) film (Figure 7f). Because of the hydrogel structure, it can easily stretch more than 1300% and has certain light transmittance (Figure 7g). Because of this special microstructure, it can also increase the voltage of the MS-TENG by seven times. When used in the data monitoring of athletes, it can also collect the biomechanical energy generated by continuous vibration during their movement through the capacitor, and provide power for small electronic equipment. The MS-TENG has been widely used in the wearable field for biological data collection or motion detection due to its good stretch and light transmittance. Wang et al. also designed a TENG based on cellulose nanofiber (CNF)/transition metal carbide and nitride (MXene) composite film for collecting human motion energy [17]. In addition, due to the introduction of carbon fiber into this TENG, the composite film has excellent mechanical strength, flexibility, and sensitivity to temperature changes.

#### 2.4.4. High-Temperature Resistance

Flexible and stretchable TENGs have always been very efficient energy solutions. However, in addition to ensuring power generation performance, this flexible equipment also needs to ensure its safety. For example, TENGs are constructed based on noncombustible materials to guarantee the safety of workers in extreme temperature conditions, but the traditional TENG cannot meet this demand. Kim et al. proposed a deformable TENG constructed by flame-retardant epoxy ion gel membrane (Figure 7a) [59]. The TENG has excellent optical transparency, flame retardancy (Figure 7b) (no combustion in the flame for 20 s), and flexibility. It can meet the safety of workers under extreme temperature conditions.

Guo et al. reported an integrated temperature-resistant and flexible hydrogel ion diode with a triboelectric nanogenerator [60]. The diode is highly rectified by a binary ethylene glycol/water solvent system and can work from −20 °C to 100 °C. A rectification ratio of up to 1201 (ethylene glycol content of 0%) and 566 (ethylene glycol concentration of 40%) was achieved, and the vibration energy can be collected by integrating the full wave rectifier circuit (with high rectification and fast operation function) with the TENG.

Li et al. synthesized ZnO nanoparticles-reinforced poly(acrylic acid)-based self-healing ion gel [61]. The ionic gel has the advantages of high-temperature resistance, high mechanical strength, and good ionic conductivity. It can be effectively used as the friction layer of the TENG, and by sandwiching the ion gel between 3 m tapes, which can be made into a high-power output TENG with a great power density of 3.15 W/m^2^.

#### 2.4.5. Self-Luminescence

Lighting is an indispensable part of wearable devices to facilitate the identification and operation in the dark and the safety of night activities, but ordinary wearable devices based on TENG can only generate instantaneous light, and the TENG also needs to power other electronic devices. Therefore, to deal with extreme darkness, Li et al., inspired by bioluminescent *Mycena chlorophos*, prepared a soft and tough elastomer luminescent composite with SrAl_2_O_4_:Eu^2+^ nanoparticles and silicone rubber as materials (Figure 7c) [62]. The screw thread steel is wrapped with composite materials, which effectively solves the problem of performance degradation of flexible triboelectric nanogenerators caused by continuous deformation, and completes the SLEN-TF. Due to the special structure of threaded steel, the SLEN-TF has high flexibility (Figure 7e). It can be made into a corresponding wearable device for motion sensing and provides great help to collect human night motion data (Figure 7d).

**Figure 7 micromachines-13-01586-f007:**
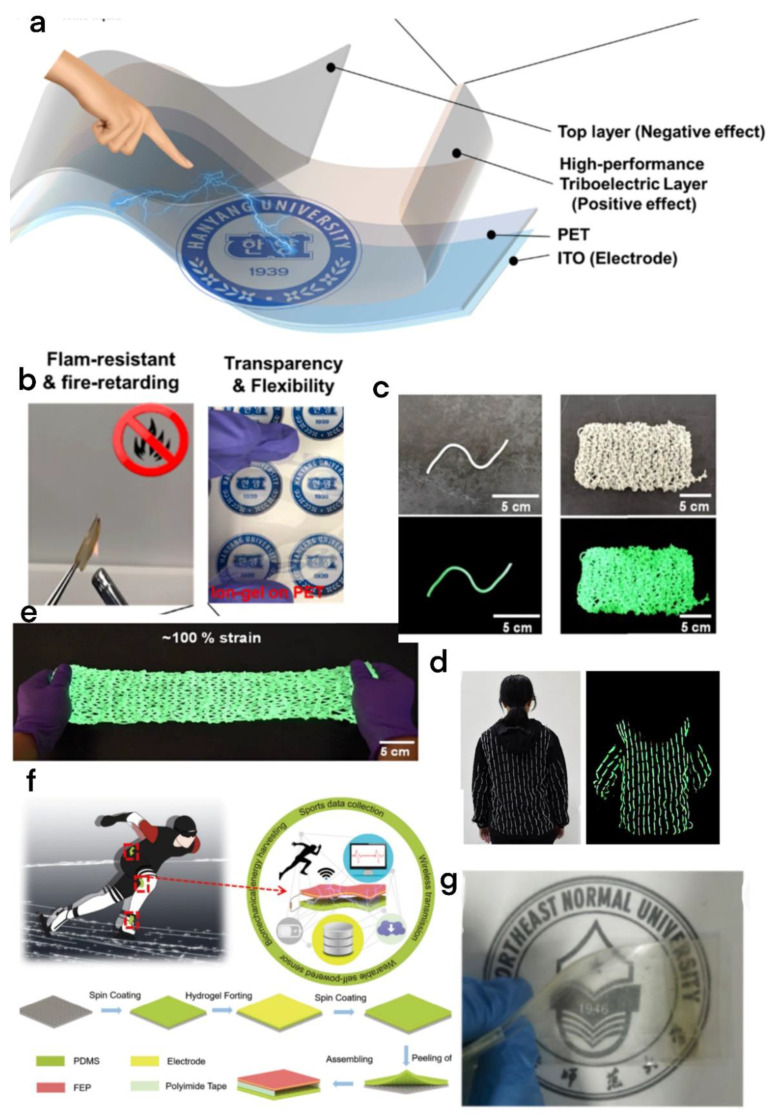
(**a**) Structural schematic diagram of the TENG system. (**b**) Flame retardancy and flexibility of the TENG device [59]. Kim et al. (2021), Elsevier. (**c**) Photos of SLEN-TF and knitted SLEN-TF-based textiles under natural light (upper) and dark (lower). (**d**) The photo of SLEN-TF textile clothing in the sun and dark. (**e**) Schematic diagram of flexibility and tensile properties of the SLEN-TF [62]. Li et al. (2022), Elsevier. (**f**) The application scene of the MS-TENG in speed skating and the manufacturing process of the MS-TENG. (**g**) The optical image of the MS-TENG in bending state [58]. Lu et al. (2022), MDPI.

### 2.5. Flexible Design toward Multifunctionality

Self-powered flexible equipment can be widely used in various fields [55,63], and multifunctionality is always highly anticipated for miniaturization, higher integration, and lower power consumption.

#### 2.5.1. Multistimuli Perception

It is similar to the skin sensor that humans cannot live without; tactile sensors play an important role in robot recognition of environmental conditions [64]. To manufacture tactile sensors similar to human skin, multistimulus analysis capabilities are essential. Rao et al. designed a tactile skin that can recognize and distinguish multiple stimuli simultaneously [65]. The skin consisted of a single electrode mode TENG (high sensitivity of 5.07 mV/Pa) and a thermal resistance electrode (combined with BiTo and RGO) (Figure 8a); the resistance temperature coefficient was 1.15%/K at 25 °C, and the range was 25–100 °C. Due to the advantages of TENG, BiTo, and RGO, the electronic skin has good flexibility (Figure 8b) and can simultaneously detect temperature and pressure.

Simultaneously, to simulate the rich sensor types of human skin, Yin et al. also reported multifunction electronic skin (CNEs) realizing contact and noncontact detection [66]. The electronic skin has a shared flexible substrate layer, which is formed by simple low-temperature hydrothermal etching of aluminum foil and has many holes on the surface (Figure 8c). The TENG (contact type) and humidity sensor (noncontact type) share the flexible substrate layer, and due to the high specific surface area brought about by the porous structure and nanosheet structure on the outer layer of the shared flexible substrate, we can easily introduce a silver nanowires (AgNWs) charged layer and a SnO_2_ moisture sensing layer on the top surface of the shared substrate (Figure 8c). CNEs provide a reliable way to construct electronic skin with complementary tactile perception and noncontact perception.

#### 2.5.2. Multi-Working Mode

To combine the excellent tensile performance with multimode electromechanical conversion, Wu et al. proposed a spiral structure fiber TENG (HS-TENG) with Ti_3_C_2_Tx as the triboelectric coating [67]. Due to its unique helical structure, HS-TENG can realize multimode electromechanical conversion including stretching, pressing, twisting, and bending, and the HS-TENG under this structure also has a 200% tensile capacity (Figure 8d). The HS-TENG, with its excellent stretching ability and multimode electromechanical conversion ability, can be widely used in electronic textiles, and it has been successfully integrated into knee pads and gloves for music player signal control through tactile sensing (Figure 8e).

Flexible electronic devices based on TENGs have good elasticity, high sensitivity, fast response, durability, and low cost, which have attracted wide interest in wearable devices and skin electronic devices, and can be effectively used in gesture recognition, motion perception, and other fields. To manufacture flexible electronic equipment with multiple working modes that can be applied to different parts of the body, Chen et al. demonstrated an ultra-thin wearable triboelectric nanogenerator (S-TENG) with coplanar electrodes [68]. The triboelectric nanogenerator has a three-layer structure, with a stretchable film of polydimethylsiloxane at the bottom and two stretchable special conductive materials (including carbon nanotubes, nano silver, and polyurethane nanofibers) embedded in the flexible PDMS film as electrodes in parallel. One of the electrodes is coated with microstructured PDMS, so it has high electronegativity and can be used as a friction layer, and the other electrode is straightforwardly used as another friction layer. Since all materials used for manufacturing the device are flexible and stretchable, the S-TENG has excellent stretchability and can be stretched in any direction, and because of the ultra-thin structure of the device, it can also adhere to the skin conformally and deform with the movement of the body. By adjusting the size of the ultra-thin structure, the S-TENG can be worn on various parts of the human body to collect motion data of the human body.

**Figure 8 micromachines-13-01586-f008:**
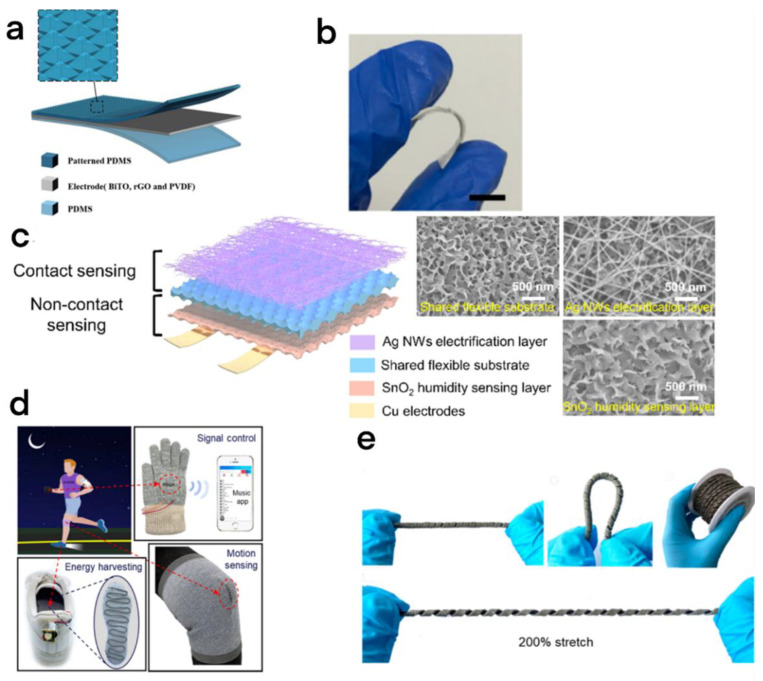
(**a**) Schematic representation of the basic structure of the tactile electronic skin. Flexible electronic textiles based on TENGs. (**b**) Electronic skin bending test [65]. Rao et al. (2020), Elsevier. (**c**) CNE schematic and SEM images for contact and noncontact sensing capabilities [66]. Yin et al. (2022), Elsevier. (**d**) Applications of sensors based on HS-TENG in signal control, body feeling, and energy collection. (**e**) Photos of the flexible HS-TENG, which can be bent and stretched to 200% tensile strain [67]. Wu et al. (2022), Elsevier.

## 3. Structural Flexibility Promotes Self-Powered Systems

### 3.1. Tactile Sensors

The self-powered tactile sensor has always been an important application field of TENGs [69], but the traditional tactile sensor based on TENGs can only detect the force in a single direction, which seriously limits its application. Wang et al. proposed a flexible and stretchable electronic skin capable of simultaneously sensing shear force and normal pressure based on a TENG (single electrode mode) (Figure 9a,b) [20]. The electronic skin is mainly composed of two layers. The upper layer includes the outermost PDMS packaging layer (for protection), the AgNW electrode shielding layer (to reduce interference and improve accuracy), and the curved PDMS layer (flexible triboelectric layer for contact and separation) with a Cu film deposited thereon. The bottom layer consists of a PDMS substrate (a triboelectric layer) and a circular electrode in the middle. The circular electrode in the middle can be changed into two identical semicircular electrodes for multidimensional force sensing.

Although the multidimensional force has dramatically enriched the application fields of tactile sensors, it is still insufficient for some special applications. Luo et al. devised an MXene/PVA hydrogel TENG (MH-TENG) [70], which uses MXene nanosheets and polyvinyl alcohol (PVA) hydrogels as raw materials and uses silicone rubber as a triboelectric layer (prevents water loss of composite hydrogel) (Figure 9c). In single-electrode mode, the measured open-circuit voltage of the MH-TENG is 230 V, and it can be extended to 200% of the original length because the relationship between the tensile length and the short-circuit voltage is monotonically increasing; it has a high sensitivity to local needlepoint stress, and can be well used for handwriting recognition and accurate sensing.

With the development of electronic skin, more and more problems have emerged. Due to the growth of bacteria and poor air permeability, long-term wearing will cause itching and even inflammation. Therefore, to improve the comfort and safety of flexible electronic skin, we need to improve its antibacterial property, flexibility, and air permeability. Shi et al. reported an electronic skin based on a flexible, breathable, and antibacterial TENG [71]. The flexible device is a self-powered electronic skin with a nanofiber network and a three-dimensional permeable layered structure created by inserting a silver (Ag NW) nanowire electrode between a thermoplastic polyurethane (TPU) detection layer and a polyvinyl hydroxide/chitosan (PVA/CS) substrate (Figure 9d). The electronic skin has excellent pressure sensitivity (0.3086 V kPa^−1^), excellent air permeability (10.32 kg m^−2^ day^−1^), and excellent antibacterial activity. It can be applied to medical sensing and the statistical analysis of sports (volleyball). Moreover, due to the excellent air permeability, stretchability (Figure 9e), and bactericidal properties of silver ions, this electronic skin can well meet the requirements of antibacterial property and comfort.

He et al. combined structural, mechanical design, microstructure modification, and transparent shielding layer coating to develop a thin, soft, and retractable self-powered tactile sensor based on the triboelectric effect for electronic skin (Figure 9f) [72]. The sensor uses sandpaper to modify the microstructure and uses a transparent silver nanowire (agent) network as the shielding layer for direct spraying, which significantly reduces the electrical crosstalk in the tactile sensor array based on the TENG. In general, this tactile sensor can recognize a wide range of pressures, as high as 0.367 mV/Pa (Figure 9g).

Cai et al. prepared a pleated PDMS/MXene composite film (pleated pattern is formed by stretching and UVO treatment) based on UV–ozone (UVO) radiation [73] and constructed a flexible TENG as a self-powered tactile sensor through the film (Figure 9h), reaching the optimal sensitivity of 0.18 V/Pa during 10–80 Pa, respectively. UVO irradiation is a simple and low-cost method for preparing the flexible self-powered tactile sensor, and this tactile sensor also has excellent sensitivity, so it has great potential in the field of fine sensing.

**Figure 9 micromachines-13-01586-f009:**
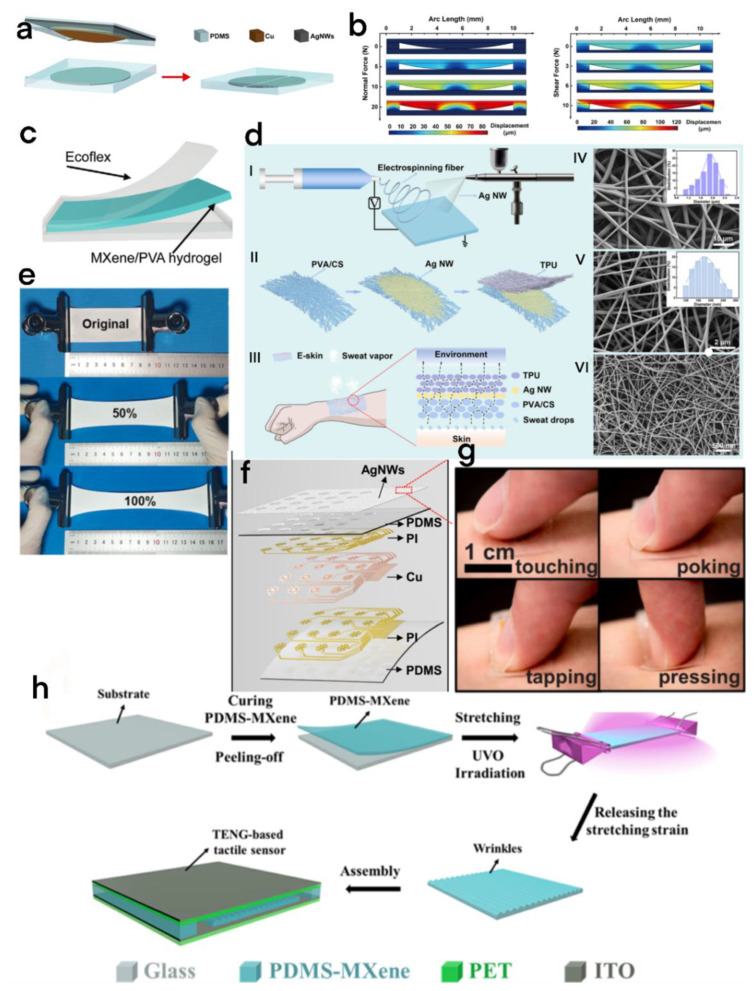
(**a**) Structural diagram of TENG-based electronic skin sensor. (**b**) The left half is the analog sensor displacement plot under the normal force of 0, 5, 10, and 20 N. The right half is the numerical calculation of the sensor’s deformation under different shear loads [69]. Wang et al. (2021), ACS. (**c**) Structural diagram of the MH-TENG [70]. Luo et al. (2021), John Wiley and Sons Inc. (**d**) Fabrication and characterization of the E-skin. (I) The fabrication process of the E-skin. (II) Schematic structure of the E-skin. (III) Schematic illustration of the transmission path in the E-skin of sweat on the arm. (IV) The SEM images of the TPU layer. The inset shows the diameter distribution. (V) The SEM images of the PVA/CS layer. The inset shows the diameter distribution. (VI) The SEM image of the Ag NW electrode layer.(**e**) Picture of electronic skin elongation test [71]. Shi et al. (2021), ACS. (**f**) Some 4 × 4 schematic diagrams of the sensor array. (**g**) Touch, poke, tap, and press with your fingers [72]. He et al. (2021), Elsevier. (**h**) Triboelectric tactile sensor based on pleated PDMS/MXene composite film [73]. Cai et al. (2021), Elsevier.

### 3.2. Display

Flexible alternating-current light-emitting (ACEL) devices, with uniform light emission and low power consumption, can be widely used in wearable devices [74,75,76]. However, the heavy power supply required to drive this kind of equipment has greatly hindered its commercialization process. Chen et al. developed a self-powered flexible ACEL device with a TENG [77], which uses a simple and low-cost corrugated aluminum electrode. The resulting voltage and current can be as high as 490V and 71.74 μA., corresponding to a most extreme momentary result power of 1.503 mW/cm^2^. To meet the power demand of ACEL, the device uses a flexible TENG, which is based on folded aluminum (AL) film. The folded structure has a fold strain percentage of 400%, which increases the effective triboelectric area and decreases the surface resistance, making it helpful for the progression of surface charges. The flexible transparent ACEL device resembles a sandwich structure and consists of an electroluminescent layer and two electrodes; the former is a composite film of ZnS: Cu and polyvinylpyrrolidone (PVP), and the latter is an electrode of indium tin oxide (ITO) and silver. Generally speaking, the successful development of self-powered flexible display has considerable application potential in human–computer interaction, artificial electronic skin, and energy-self-sufficient communication in the IoTs.

Sun et al. proposed a flexible transparent self-powered display, which is characterized by combining ACEL devices with a TENG and is a display system sharing transparent electrodes (Figure 10a) [78]. First, the emission layer of ZnS: Cu @ PDMS was sandwiched between the transparent electrode of a single-walled carbon nanotube (SWCNT) and ITO/PET substrate to prepare transparent ACEL. Then, they assembled ACEL with two transparent triboelectric layers, ITO and PDMS. This device has a voltage of 200 V and a short-out current of 6 μA. The ACEL device can be easily lit, and the integrated system has a light transmittance of more than 80%.

Shan and his team introduced MXene into the emission layer to prepare a new self-powered ACEL device (Figure 10b,c) [79]. Among them, MXene is a new two-dimensional (2D) material composed of transition metals and carbides/nitrides (mainly made by eroding the max phase), which have excellent chemical properties [80]. PDMS is an important material of TENGs, and its relative dielectric constant will improve by the load of MXene to varying degrees. Therefore, when 0.25 wt% MXene was loaded in the emission layer, the brightness of ACEL was also increased by 500%. The TENG in the system is a contact mode, with Cu as a positive charge collector and PTFE as a negative charge collector to continuously provide power for ACEL equipment.

**Figure 10 micromachines-13-01586-f010:**
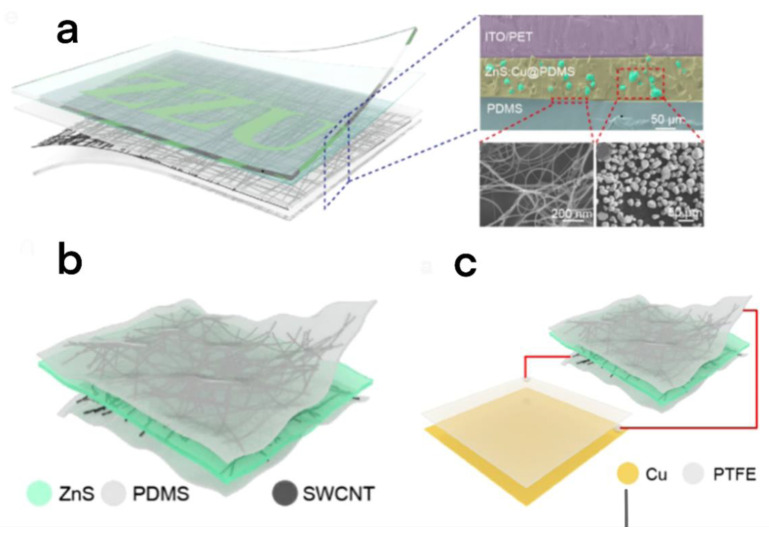
Structure of flexible self-powered display screen. (**a**) Structural diagram of an integrated self-powered transparent ACEL system [78]. Sun et al. (2022), Elsevier. (**b**) Schematic diagram of ACEL device, consisting of two SWCNT transparent electrodes and ZnS: Cu emission layer. (**c**) Conceptual diagram of self-powered ACEL device [79]. Sun et al. (2021), Elsevier.

A flexible self-powered display screen has great potential in the fields of IoT and personalized display devices due to its special structure and self-powered characteristics. However, the single black and white color tone greatly limits its commercialization. To solve this problem, Chang et al. integrated TENG with stretchable multicolor AC electroluminescent (ACEL) devices, and a new self-powered and self-repairing multicolor display was proposed (Figure 11a,d) [81]. This self-powered self-healing multicolor display has a structure with high stretchability and self-healing efficiency by adding reversible dynamic imine bonds to ordinary PDMS (reaching 2500% tensile performance and 96% self-recuperating effectiveness at room temperature); through this special structure (Figure 11c) as electrode substrate and emitter layer matrix, they successfully manufactured naturally stretchable and self-mending ACEL (SSH/ACEL) devices (Figure 11b).

### 3.3. Medical Devices

With the gradual development of the medical field, we have greatly expanded the application scenarios of medical devices. However, the common rigid structure and DC power supply severely limit the application scenarios of the device, while the flexible sensor has good application prospects in personal movement detection, health monitoring of various parts of the body, analysis of data for medical diagnosis, and intelligent human–machine interaction because of its great response to various types of external stimuli [82,83,84,85,86]. Therefore, to enrich the application of medical equipment, flexible, stretchable, and self-powered equipment is the future development direction [87].

Liu et al. designed a flexible self-powered drug release device (FDRD) which has the benefits of self-powered, flexible structure, and controllable delivery [87]. This device is an integrated TENG, including three TENG units. Sufficient power output can be generated through mutual assistance between the three units; each TENG unit is made out of an acrylic plate (improve protection and support), fluorinated ethylene propylene (FEP) film clung to a copper electrode, a copper film, a sponge (provide soft layer), and a kapton film (Figure 12b,c). In this device, the FEP film and copper film serve as the triboelectric layer, the copper film also serves as the electrode, the kapton film (thickness 200 µm) serves as the support layer, and the acrylic plate and sponge serve as the card base and the soft layer. The TENG can effectively obtain energy from biomechanical energy (Figure 12a), and due to the stable voltage result of the TENG, the device can record the ultraviolet–visible absorption spectra of little molecules (which have methylene blue, sodium fluorescein, and Rhodamine 6G) let out of FDRD in real time, and since the poly (3-hexylthiophene) (P3HT) film has electronic/ionic conductivity and solution processability in Na_2_SO_4_ aqueous solution, the device can adjust the special switchable wettability by turning the mechanical switch on and off.

Human sweat contains hormones (for regulation), metabolites (to be discharged to the body), electrolytes, and amino acids (important components of protein). These substances assume a significant part in reporting health status and can be used for medical diagnosis of skin diseases, diabetes, and other diseases. However, the early sweat sensor is a rigid base, and it will feel uncomfortable after wearing it for a long time. Therefore, light, soft, flexible electronic devices are more and more popular, and because the soft and stretchable device has a larger contact area with the skin, the data obtained in sweat monitoring are also more convincing. To meet the light and flexible characteristics of the sweat sensor, Qin and his team developed a fully flexible, self-powered sweat sensor that was based on conductive cellulose hydrogel (Figure 12d) [88]. Due to the use of special cellulose nanocomposites, the hydrogel has a tensile property of 1530% and a self-healing rate of 95%, and high conductivity within 10 s without external stimulation. This cellulose nanocomposite contains oxidized CNF (TOCNF), polyaniline (urease sensor), tempo, and PVA/BORAX (PVAB). The TOCNF/PANI-PVAB hydrogel (CCPPH) electrode is encapsulated by a flexible and wear-resistant PDMS and assembled with an ion-selective membrane (ISM) capable of detecting the concentrations of Na^+^, K^+^, and Ca_2_^+^. The ISM and PDMS layers encapsulate the flexible electrode and act as a positive triboelectric layer and a negative triboelectric layer, respectively.

Meanwhile, Li et al. proposed a bionic sweat-resistant and wear-resistant friction electric nanogenerator (BSRW-TENG) for medical data detection during exercise (Figure 12e) [89]. Since the two friction layers of the BSRW-TENG replicate the micro/nanostructure of the lotus leaf surface, the two friction layers of the BSRW-TENG (raw materials are elastic resin and PDMS) have superhydrophobic and self-cleaning capabilities, and this special micro/nanostructure can also effectively improve the power-generation capacity of the TENG. In addition, with the support of this superhydrophobic structure, the BSRW-TENG has excellent antiperspiration ability and it can be worn on all parts of the human body for a long time for medical data analysis.

**Figure 12 micromachines-13-01586-f012:**
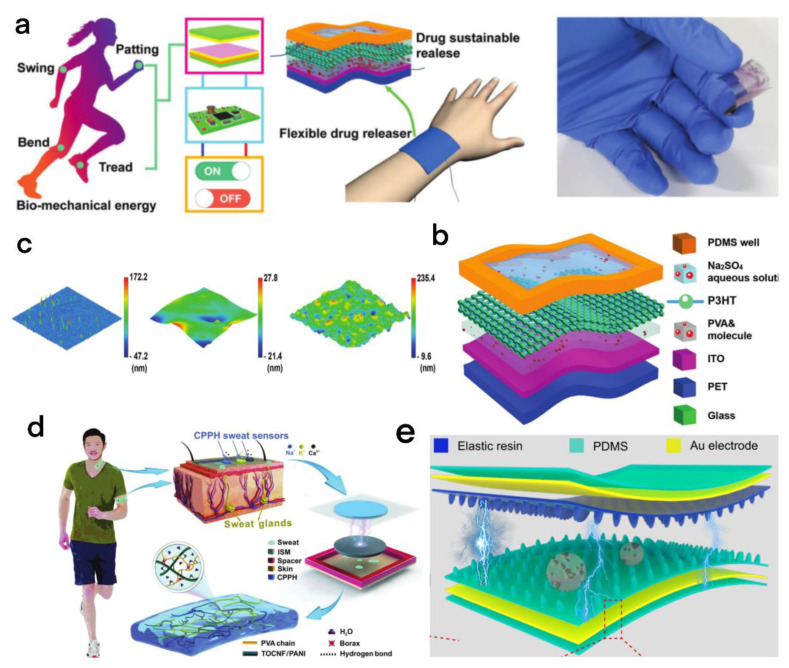
(**a**) The TENG collects biomechanical energy to provide power for the FDRD. (**b**) Structure of the FDRD. (**c**) The surface morphology of the ITO layer, the PVA molecular layer, and the P3HT layer was recorded with an atomic force microscope (AFM) [87]. Liu et al. (2020), John Wiley and Sons Inc. (**d**) Cellulosic conductive hydrogel for self-powered sweat measurement, CPPH electrode material, microstructure diagram, and CPPH sweat sensor module structural diagram. During body movement, the sweat sensor monitors the concentration of sweat ions in real time. The CPP sweat sensor is placed on the skin’s sweat glands to detect and quantify Na^+^, K^+^, and Ca^2+^ [88]. Qin et al. (2022), John Wiley and Sons Inc. (**e**) Schematic structure of the BSRW-TENG [89]. Li et al. (2022), Elsevier.

Electrical stimulation has always been a good method to treat the loss of muscle function. To stimulate muscles for a long time, Wang et al. reported a flexible wearable self-powered muscle stimulation device (Figure 13a) [90]. The equipment adopts a stacked TENG structure that can be well integrated into human movement. This structure first folds the PET sheet into a “zigzag” structure, and stores the mechanical deformation energy in the form of elastic properties (this special “zigzag” structure can automatically restore the TENGs after each compression to the original position (Figure 13a)), and then attaches the aluminum film to each surface of the folded PET film, as an electrode for discharging charge. When this structure deforms under the applied pressure, charge transfer will occur, and mechanical energy will be converted into electrical energy.

The wound healing can be accelerated by receiving appropriate electrical stimulation, but the bloated power supply system limits the use scenario of this method. To remove the bloated power supply system, Du et al. fabricated a single-terminal TENG-based skin electronic patch by embodying the conductive and photothermal composite hydrogel of polypyrrole (PPy) and watery F127 in elastic silicone [91]. The dermal patch has excellent extensibility (~300%), and biocompatibility, motion induction, and promotion of wound healing were achieved through synergistic use of electrical stimulation and photothermal heating capacity (Figure 13b,c). In addition, the electromechanical signal conversion that occurs due to stretching or deformation of wearable TENG e-skin can effectively compensate for the loss of cognition (squeezing and touching) caused by the wound.

**Figure 13 micromachines-13-01586-f013:**
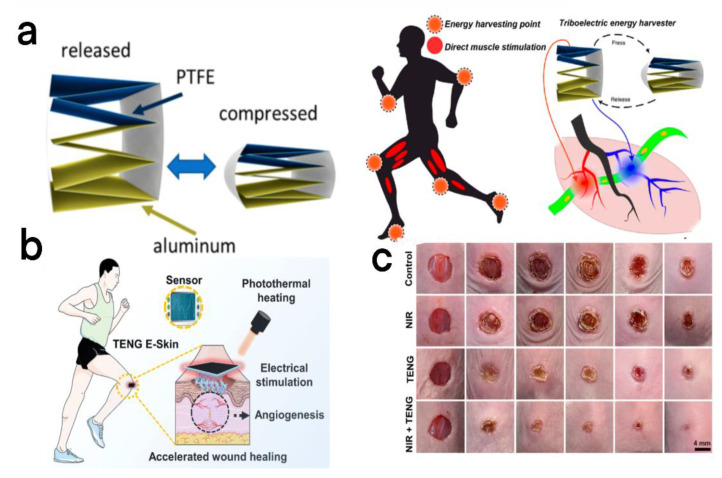
(**a**) The left half shows the tensile and compressive properties of the TENG, and the right half shows the concept of muscle electrical stimulation directly powered by the TENG [90]. Wang et al. (2019), ACS. (**b**) Functions of wearable TENG e-skin patch in the wound: near-infrared photothermal heating, electrical stimulation, and sense. (**c**) Skin wound images were taken on days 0, 3, 5, 7, 9, and 11 of the control group, NIR, TENG, and NIR + TENG [91]. Du et al. (2022), Elsevier.

### 3.4. Cathodic Protection Device

In the face of metal corrosion [92], the traditional cathodic protection device cannot output current stably for a long time and has the characteristics of high cost, large volume, and narrow application range [93,94]. The difference is that the flexible TENG can obtain available energy from raindrops, waves [95], and a series of small vibrations. Moreover, due to the special flexible structure of the TENG, it can cling to all places where it is convenient to obtain energy and continuously and stably increase the current for the cathodic protection system. Therefore, in the anticorrosion application of ships, giant cranes, and other heavy metal equipment, connecting the metal to the self-powered cathodic protection system operated by the TENG can effectively protect the metal from corrosion.

Cai et al. prepared a flexible cellulose/collagen/graphene oxide thin film TENG (GO CC TENG) (Figure 14a) [23]. Under the external mechanical energy movement frequency of 1 Hz and the load resistance of 400 mΩ, the maximum power of the GO CC TENG reached 196 μW, which can be effectively applied to cathodic protection. Among them, graphene oxide (GO) used as raw material has the advantages of extremely high specific surface area and [96] extremely significant mechanical properties. Collagen is biocompatible and thermally stable within a certain temperature range. Cellulose has the advantages of biodegradability, nontoxicity, and low cost.

The self-sufficient cathodic protection system powered by the TENG is usually used in harsh environments such as ships. Liu et al. prepared hybrid coatings with self-healing hydrophobicity through micro-arc oxidation (MAO) and fluorinated sol–gel (FSG) coatings (Figure 14b) [97]. The hybrid coating is used as a triboelectrification layer in a self-powered cathodic protection system, and if the fluorine composition on the coating surface is damaged, the system will transfer the loaded perfluoro silane to the damaged surface to achieve self-healing. This self-powered cathodic protection system with hydrophobicity and self-healing ability has great application potential in the field of rust removal in harsh environments.

At the same time, Sun et al. obtained a self-healing silicone-modified polyurethane (PU) coating through tin-catalyzed hydroxypropyl silicone oil and hexamethylene diisocyanate (Figure 14c,d) [98]. This layer is used as the triboelectric layer of the TENG. When the friction layer is worn during the energy-collection process, the broken organosilicon-modified polyurethane polymer chain will be gradually crosslinked through hydrogen bonding within 30 min to achieve a self-healing effect. This kind of TENG with a self-healing function can be widely used in a self-powered cathodic protection system to improve the stable anticorrosion function of metal equipment.

Wang and his team introduced a wind-powered flexible TENG as a power supply for a self-powered cathodic protection (CP) system (Figure 14e) [99]. The system uses hydrophobic polytetrafluoroethylene (PTFE) flexible film as a friction layer and generates an electric charge together with polyvinyl alcohol (PVA) film, which is continuously transferred to the cathodic protection system, reaching a power of 1.74 mW. Usually, the system can effectively convert wind energy to prevent metal corrosion in marine facilities.

**Figure 14 micromachines-13-01586-f014:**
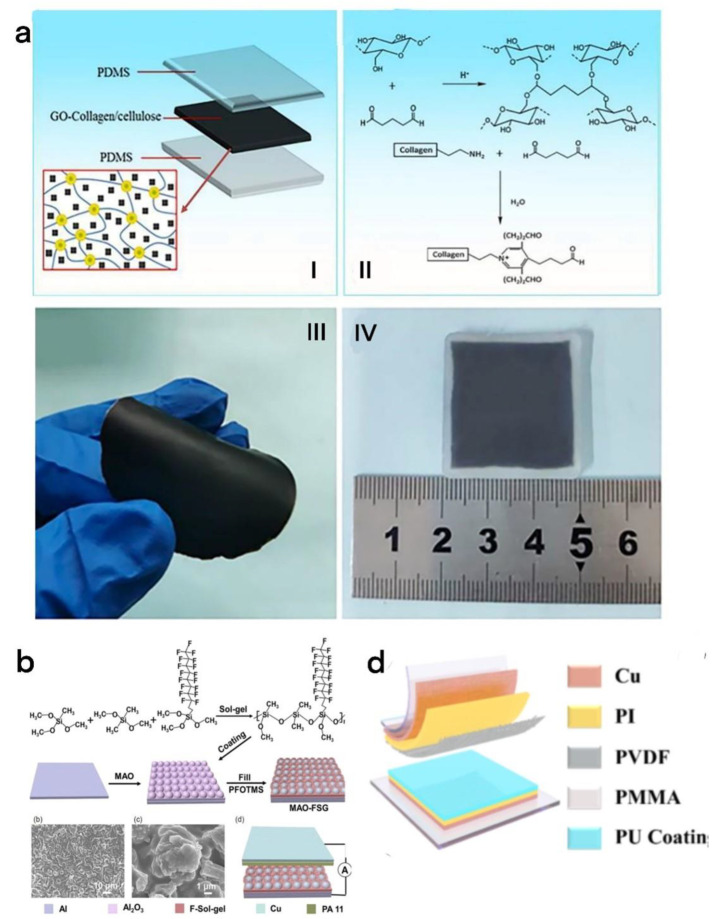
(**a**) (I) the structure diagram of the GO-CC-TENG. (II)the manufacturing diagram of the GO-CC-TENG. (III), (IV) the picture of the synthetic go-collagen/cellulose membrane and the GO-CC-TENG [23]. Liu et al. (2022), Elsevier. (**b**) The upper part is the schematic diagram of the manufacturing process of the Mao/FSG coating, and the lower part is the SEM image of the Mao/FSG coating and the schematic diagram of the MF-TENG [97]. Liu et al. (2022), CNKI. (**c**) Schematic representation of the molecular level repair mechanism of linear silicone-modified polyurethane. (**d**) Schematic diagram of the PU-based linear silicone-modified self-healing TENG [98]. Sun et al. (2022), ACS. (**e**) Schematic diagram and mechanism diagram of the PTFE-based flexible TENG for collecting wind energy and moisture resistance [99]. Sun et al. (2021), Elsevier.

## 4. Conclusions and Perspectives

In general, current trends in the adaptable design of TENG-based self-powered systems were concisely introduced, and strategies to achieve high-performance energy conversion, high-sensitivity detection of mechanical stimulation, and reliable output in harsh environments were systemically reviewed. Cutting-edge progress of self-powered systems with multifunctionality in optoelectronics, medical treatment, sensing, and cathodic protection was discussed. With the structural flexibility, a TENG-based electronic device can not only change the shape, for instance, being interwoven into textiles, and perfectly fit on the skin, but can also change the raw materials to achieve the effects of ventilation, sterilization, physical and chemical stability, and flame retardancy for harsh environments. To date, wide applications of structural flexible TENGs in biosensors, human–machine interaction, and intelligent robots have been explored, either as sustainable power sources or as self-powered sensors themselves to help collect environmental or human motion data.

Nevertheless, challenges and barriers to the practical application of TENGs in the current situation also need to be addressed. Some issues are commonly faced by TENGs, and some may be individual due to the integration of TENGs with other electronics to form self-powered multifunctional systems.

### 4.1. Increase Output Power

There are currently three main methods to improve the device’s power output: changing the triboelectric layer’s material, modifying the triboelectric layer’s surface, and changing the structure of the TENG.

Surface modification of the triboelectric layer includes chemical and physical methods. The chemical method is mainly grafting some chemical groups to generate or accept electrons. Physical transformation methods include modeling, plasma processing, and engraving. In general, the output power can be improved through an appropriate combination of physical and chemical methods.

For the material selection of the triboelectric layer, materials are always expected to be more triboelectrically negative, if other physicochemical properties can be guaranteed, and the exploring of new triboelectric materials is an everlasting task for enhancing the output of TENGs.

The structure of the TENG is also an essential factor affecting output efficiency. A major future research direction is how to design the structure to make it more integrated with the human body and more consistent with the physical rules while maintaining stable and powerful performance.

### 4.2. Higher Sensitivity

TENG-based sensors are suitable to be embedded in or integrated with the Internet of Things and intelligent robots if improved sensitivity, reduced interference, and customized structure can be obtained.

For high-precision selection, we should try our best to find highly sensitive materials. Of course, we should synthesize and develop new materials. In order to reduce interference, we can design multiple groups of filter circuits in future research. Structural customization is also a major reason that restricts the accuracy of sensors. Therefore, we can devote ourselves to developing higher-precision printers, finding high-precision materials suitable for printing, or achieving high customization through electrospinning and laser etching.

### 4.3. Comfortability and Stability

Comfort and stability have always been the parameters that people pay attention to. Due to the working principle of the TENG, long-time triboelectric between layers is required. Therefore, to reduce the loss and prolong the TENG’s service life and stability, future research can focus on finding more wear-resistant materials or a triboelectric method with a low loss rate.

In addition, stability in the unique working environment is also one of the factors that must be considered, such as high temperature and humid climate. In the future, we can choose high-temperature-resistant materials and modify the surface or integrate the hydrophobic membrane into TENGs. As sense is the main field of flexible devices, various minor disturbances will inevitably occur in sensing, which will seriously affect the stability of devices. Therefore, in the future, we can focus on circuit design and achieve the purpose of reducing or eliminating disturbances through special filter circuits.

To improve the user experience, comfort and stability are indispensable. Since the flexible equipment itself has a certain degree of flexibility and stretchability, the emphasis can be placed on the air permeability and weight of the material, and the future research direction can be set on finding more advanced materials, such as cotton, silk, linen, chiffon, cotton silk, etc. Alternatively, in the future, we can also focus on the textile method of electronic textiles. Flexible wearable electronic devices made by unique textile methods can also have good air permeability and are lightweight.

### 4.4. Environmental Tolerance

Improving environmental tolerance is the key to enhancing the stability and safety of TENGs. Since it is difficult for traditional equipment to collect energy under extreme conditions, such as high-temperature conditions, acid–base conditions, etc., and the TENG is a rare multichannel energy-collection device, it proposes higher requirements for the environmental tolerance of TENGs. For high-temperature problems, future research can focus on finding more high-temperature-resistant materials or adding auxiliary heat-dissipation devices to solve the problem of high-temperature resistance. For acid–base issues, in addition to finding more acid–base-resistant materials in the future, the superhydrophobic surface can also be prepared by surface modification and other methods. At the same time, the TENG also faces extreme conditions such as stretching and use in the dark. We can focus on finding more elastic materials, adding LED lamps for auxiliary lighting, or adding biological luminescent enzymes to solve this problem.

### 4.5. Multifunctionality

Increasing the function of flexible TENG equipment is also a significant direction to improving user experience. For example, multilayer flexible self-powered equipment with blood glucose detection and motion detection capabilities needs to integrate the multilayer films, and the wearer needs to generate enough motion to provide energy, which has high requirements on the toughness, thickness, electronegativity, and functionality of the films. In this case, the future should focus on the method of integrating films with different functions so that the films with different functions can play their respective roles, and the output power of the TENG should be increased so that it can support more functions.

### 4.6. Cost Reduction

Low cost is an intrinsic advantage of TENGs in comparison with other energy harvesting technologies, especially for the scavenging of waste mechanical energy from the ambient environment. However, seeking lower costs always conforms to the keep-moving rule. As far as the adaptive design of TENGs for applications in IoT, the raw material cost can be conveniently reduced by selecting inexpensive nylon films (nm) or even waste/natural materials as the main raw material under the precondition of satisfactory functionality.

## Figures and Tables

**Figure 1 micromachines-13-01586-f001:**
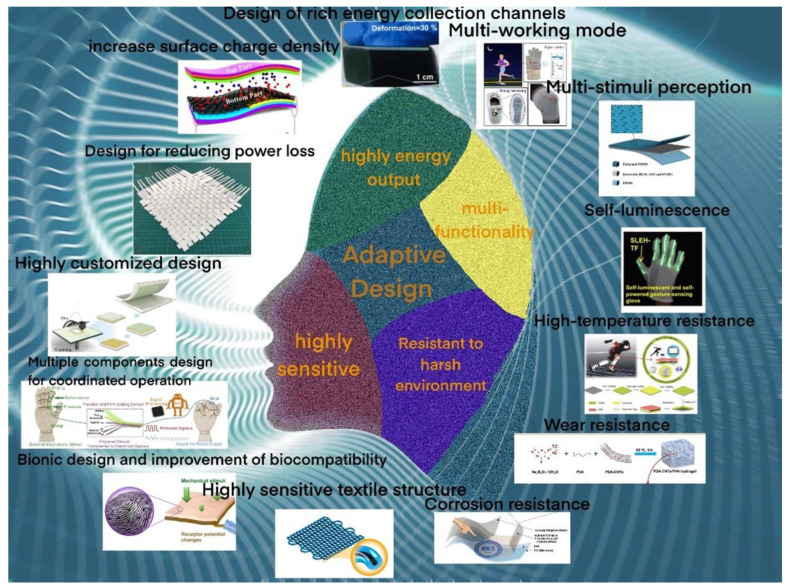
Adaptive design of self-powered systems.

**Figure 2 micromachines-13-01586-f002:**
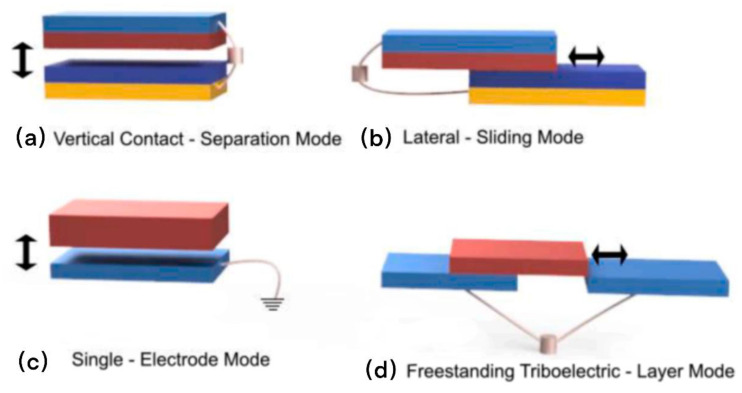
(**a**–**d**) Four working principles of the TENG [29] (Ryan et al. (2021), Elsevier).

**Figure 5 micromachines-13-01586-f005:**
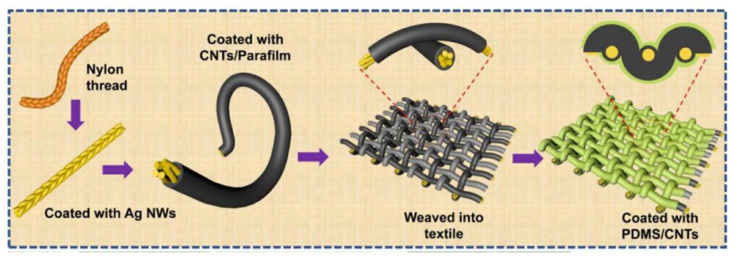
Principle and manufacturing flow chart of piezoresistive fiber triboelectric sensor (PRF-TES) [52]. Yan et al. (2022), Elsevier.

**Figure 11 micromachines-13-01586-f011:**
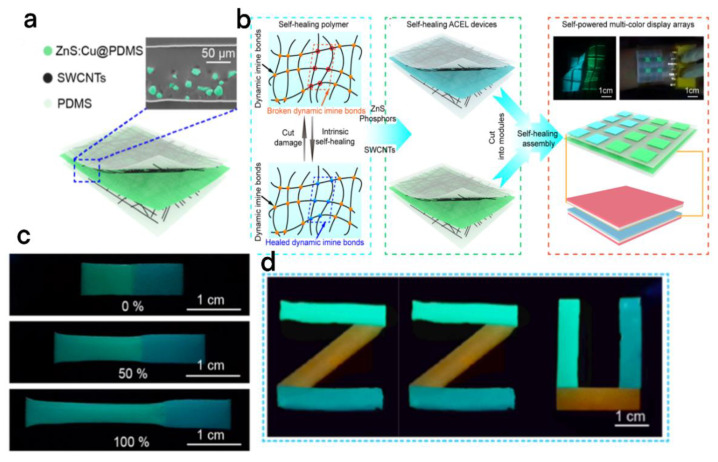
(**a**) Schematic representation of the stretchable self-healing ACEL device consisting of two electrodes and an emissive layer. (**b**) The working mechanism of the self-healing polymer and the manufacturing route of the ACEL self-powered red multicolor display. (**c**) Photographs of the healed emissive layer at different loads (0%, 50%, and 100%). (**d**) Multicolor display with a ZZU [81]. Chang et al. (2022), Elsevier.

## Data Availability

Data available on request from the authors.

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
