# Peer review of "Structural Flexibility in Triboelectric Nanogenerators: A Review on the Adaptive Design for Self-Powered Systems"

_micromachines, 2022, doi:10.3390/mi13101586_

Round 1

Reviewer 1 Report

This is a fine review article manuscript. The authors have tried to review the topic extensively and the manuscript seems overall sound. The references are also mostly recent.

However, there are minor issues which needs to be addressed.

1. The conclusion is well written, however the introduction requires some more details. Piezoelectric nanogenerators (PENGs) came into the foray before the triboelectric nanogenerators (TENGs), however there is no apt mention of PENGs in the introduction. The advantages of the TENGs over PENGs may be elaborated to address this issue.

The authors may refer to the following references.

https://doi.org/10.1016/j.jallcom.2022.164060

10.1109/TNANO.2016.2520019

https://doi.org/10.1039/C9RA10811D

https://doi.org/10.1039/C8NJ04751K

https://doi.org/10.1002/slct.201602046

https://doi.org/10.1002/aenm.201601016   https://doi.org/10.1039/C5NR02067K   https://doi.org/10.1016/j.nanoen.2021.105789   https://doi.org/10.3390/nano11071712  

2. As this article is about nanogenerators it is necessary to mention ZnO NWs briefly in the introduction.

 https://doi.org/10.3390/nanoenergyadv2020008

https://doi.org/10.3390/nano11061430

http://dx.doi.org/10.1088/2053-1591/2/10/105017

3. In section 2.1 in the manuscript, the schematic of the working mechanisms of the different modes of the operation of the TENGs are required.

4. Theoretical science/descriptions of the origin of the triboelectric effect if available in the literature needs to be added appropriately.

5. There are some minor issues in the manuscript. There is no page number, please add it. There are some random capital letters, full stops at random places in the manuscript. Also, subscripts and superscripts are also not addressed. 

Author Response

Response to reviewers

micromachines-1916952

Title: Structural Flexibility in Triboelectric Nanogenerators: A Review on the Adaptive Design for Self-Powered Systems

Dear Editor,

We appreciate the reviewers’ comments, which are very helpful for the improvement of the presentation of our paper. We have revised our paper according to their comments. Replies to each referee are provided.  

Thanks for your consideration and we wish it could meet the criteria of the publication.

Sincerely,

Ning Wang

+++++++++++++++++++++++++++++++++++++++++++++++++++++++++++++

Reviewer #1: Comments:

  1. The conclusion is well written, however the introduction requires some more details. Piezoelectric nanogenerators (PENGs) came into the foray before the triboelectric nanogenerators (TENGs), however there is no apt mention of PENGs in the introduction. The advantages of the TENGs over PENGs may be elaborated to address this issue.

Response: Thanks for your professional review work on our manuscript. In the introduction, I appropriately added relevant content about PENG Discussed the relevant contents from the discovery of ZNO in 2001 to the emergence of PENG in 2006 and the invention of TENG in 2012

  1. As this article is about nanogenerators it is necessary to mention ZnO NWs briefly in the introduction.

Response: We really appreciate your valuable comments. According to your good suggestions, In the introduction, we supplemented the discovery process related to ZNO and further elaborated its relationship with PENG

  1. In section 2.1 in the manuscript, the schematic of the working mechanisms of the different modes of the operation of the TENGs are required.Response:

Response: Appreciate your valuable comments I have supplemented the relevant schematics of TENG's four working modes in 2.1, and submitted the corresponding copyright certificates

  1. Theoretical science/descriptions of the origin of the triboelectric effect if available in the literature needs to be added appropriately.

Response: Thank you for your comments. I supplemented the principle of the triboelectric effect in 2.1, and elaborated the triboelectric effect due to the different gain and loss capacities of different atoms

  1. There are some minor issues in the manuscript. There is no page number, please add it. There are some random capital letters, full stops at random places in the manuscript. Also, subscripts and superscripts are also not addressed. 

Response: Thanks for your professional review work on our manuscript. I added a page number, correction of letter capitalization error and modified superscript and subscript problems. The modified positions are marked in the text

Reviewer 2 Report

In this review, Zhao and co-workers sums up the recent achievements in the fields of self-powered systems. This review is well written. Therefore, I recommend accepting it after minor revision.

Comments are given as follows:

1. Please add the Table in the main text, to make a comparison of the different types of TENGs, such as output power density, output current, etc.

2. Some related Ref. are suggested to cite, Nano Energy, 88, (2021), 106256; Nano Energy, 90, (2021), 106536.

Author Response

Dear Reviewers:

    We very appreciate your valuable comments, which are all valuable and very helpful for revising and improving our paper, as well as the important guiding significance to our researches. We have studied comments carefully and have made corrections which we hope meet with approval. The corrections are marked in red in the revised manuscript.

We will appreciate for all your advice, if you have any question, please do not hesitate to let us know.

Best Regards,

Sincerely yours

Ning Wang

+++++++++++++++++++++++++++++++++++++++++++++++++++++++++++++

Reviewer #2: Comments:

  1. Please add the Table in the main text, to make a comparison of the different types of TENGs, such as output power density, output current, etc.

Response: Thanks for your professional review work on our manuscript. Due to the lack of sufficient examples of nanogenerator power, the lack of comparability between the examples listed in this paper, and the disunity in power description (such as maximum power instantaneous power effective power), I haven't found a better way to describe the form

  1. Some related Ref. are suggested to cite, Nano Energy, 88, (2021), 106256; Nano Energy, 90, (2021), 106536.

Response: Thank you for your comments. It makes my understanding of ZNO more profound. I have applied the reference you recommended to [19]

Reviewer 3 Report

In this review, the authors organized various researches related to the development of structural flexible triboelectric nanogenerators for self-powered systems. They summarized numerous relevant researches and gave a direction for the future development. However, there are some points to check and revise for improving the clarity and readability of manuscript. To improve the quality of manuscript, please check and revise based on the list below.

1. First of all, I noticed that there is no statement related to copyright in figure captions. Please get the permission for reprint from publishers and add statement, for example, "Adapted with permission from [ref #]. Copyright © 2022 Elsevier." Please check other review articles published in Micromachines.

2. There are subscript and superscript errors in chemical notation and units, for example, Al2O3 and WM-2. Please carefully check the subscripts and superscripts in manuscript.

3. In Figures 2h and 2i, authors showed a schematic of fabrication process or assembly process. However, in relevant contents, there is no explanation of fabrication process. In my opinion, schematics of finalized device are enough as displayed in Figure 2a. Please consider the modification of Figures 2h and 2i.

4. I think the description of Figure 3h is not intended by authors. Please check the figure caption of Figure 3h.

5. In my opinion, the authors need to adjust the size of figure and text to improve the readability. It looks like authors put the original figures together without modifications. Therefore, it is hard to see the text in figures.  

Author Response

Dear Reviewers:

    We very appreciate your valuable comments, which are all valuable and very helpful for revising and improving our paper, as well as the important guiding significance to our researches. We have studied comments carefully and have made corrections which we hope meet with approval. The corrections are marked in red in the revised manuscript.

We will appreciate for all your advice, if you have any question, please do not hesitate to let us know.

Best Regards,

Sincerely yours

Ning Wang

+++++++++++++++++++++++++++++++++++++++++++++++++++++++++++++

Reviewer #3: Comments:

  1. First of all, I noticed that there is no statement related to copyright in figure captions. Please get the permission for reprint from publishers and add statement, for example, "Adapted with permission from [ref #]. Copyright © 2022 Elsevier." Please check other review articles published in Micromachines.

Response: Thanks for your professional review work on our manuscript. I have added the corresponding copyright notice after the pictures

  1. There are subscript and superscript errors in chemical notation and units, for example, Al2O3 and WM-2. Please carefully check the subscripts and superscripts in manuscript.

Response: Appreciate your valuable commentsI carefully checked the superscript and subscript problems in the manuscript and corrected them

  1. In Figures 2h and 2i, authors showed a schematic of fabrication process or assembly process. However, in relevant contents, there is no explanation of fabrication process. In my opinion, schematics of finalized device are enough as displayed in Figure 2a. Please consider the modification of Figures 2h and 2i.

Response: Thank you for your comments.I deleted Figure 2h and Figure 2i and modified the corresponding layout. Since a new picture has been added, Figure 2 has become Figure 3

  1. I think the description of Figure 3h is not intended by authors. Please check the figure caption of Figure 3h.

Response: We really appreciate your valuable comments. Due to my negligence, I made a mistake in 3h picture explanation. I modified it. Now its picture number is 4h

  1. In my opinion, the authors need to adjust the size of figure and text to improve the readability. It looks like authors put the original figures together without modifications. Therefore, it is hard to see the text in figures.  

Response: Appreciate your valuable comments I have revised the typesetting of Figures 3, 7, 8, 9, 12 and 14 and the content of the picture has been appropriately deleted for better viewing

Round 2

Reviewer 3 Report

The authors fully reflected on the comments. I recommended accepting this paper for publication in Micromachines.

However, there are some errors in notation, for example, WM^(-2) on page number 12. The authors should carefully check any notation in this review article.